# Preparation of Special Wettability Quartz Sand Filter Media and Its Synchronous Oil/Water Mixture Separation and Dye Adsorption

**Che Yinglong [1], Guo Hanyue [1,2], Man Shide [1], Zhang Tingting [1,3,*] and Wei Bigui [1,3,*]**

1    School of Environmental and Municipal Engineering, Lanzhou Jiaotong University, Lanzhou 730070, China
2    Jiangxi Architectural Design and Research Institute Group Co., Ltd., Nanchang 330046, China
3    Key Laboratory of Yellow River Water Environment in Gansu Province, Lanzhou 730070, China
*    Correspondence: zhangting1229ztt@163.com (Z.T.); weibg@mail.lzjtu.cn (W.B.)

**Abstract:** To efficiently and synchronously separate oil/water mixture and adsorbed dyes, corn-cob-covered quartz sand (CCQS) filter media with underwater superoleophobic qualities and underoil extremely hydrophobic qualities were fabricated by grafting a corn cob onto the surface of quartz sand using the dip-coating technique. Due to the introduction of more hydrogen bonds on the quartz surface and the construction of a rough structure, the underwater oil contact angles and underoil water contact angles of the CCQS were 150.3~154.6° and 132.2°~154.6°, respectively. A separator for oil/water separation was devised, and the CCQS-filled separator could synchronously separate the oil/water mixture and adsorb malachite green. The separation efficiency of the oil/water mixture was over 99.93%, the removal rate of MG was 99.73%, and the adsorption capacity was 7.28 mg/g. The CCQS could keep its wettability steady under challenging environmental circumstances. Therefore, the study offered a novel concept for the successful oil/water mixture separation, while synchronously adsorbing dye.

**Keywords:** quartz sand filter media; underwater superoleophobic; extremely hydrophobic underoil; dye adsorption; oil/water separation

## 1. Introduction

The growing demand for various oils and increasingly more oily wastewaters were caused by the development of petroleum, machinery manufacturing, transportation, textile, catering, and other industries, with the oily wastewater disposal entering the water [1]. Every year, large amounts of oil are released into the sea, posing a major danger to marine creatures [2]. China has a high demand for dyes because of the demand for textile, printing, dyeing, leather, and other industries, so a lot of dye effluent is released into bodies of water each year [3,4]. Dye wastewater has the characteristics of high chromaticity, high toxicity, various components, nonbiodegradable, difficult to remove, etc., which cause great harm to animals, plants, and even human life when discharged into water. At present, oil wastewater and dye wastewater are removed separately in industry, and there are few reports focusing on the method of synchronous removal.

Special wettability materials exhibit various wetting behaviors in both water and oil, which mainly include superoleophilic and superhydrophobic materials and super-hydrophilic and superoleophobic materials, and dye adsorption can take advantage of hydrophilic groups [5–10]. Wettability surfaces are of great significance in antifog, anti-icing, responsive switch, oil and water separation, high-load water equipment, nondestructive liquid transfer, directional liquid transport, blood-compatible materials, and other fields [11–14]. Micro/nanoscale surface features and the chemical makeup of the surface influence its wettability [15–18]. Substances introduced onto the material surfaces mainly include chitosan, $SiO_2$, $Cu_2S$, $TiO_2/CuO$, hydroxyl groups, biomass, hydrophilic polymers,

ester groups, etc. [19–25]. Li et al. [26] used walnut shell powder as the base material, and the removal effectiveness of various oil/water mixtures was over 99.94%. Jiang et al. [27] created a superhydrophobic and superoleophilic stainless steel mesh film to efficiently separate oil/water mixtures by using hydrophobic group self-assembly and covalent grafting to create hierarchical micro/nanostructured surfaces. Wu et al. [28] bonded $Fe_3O_4$ nanoparticles with sponges through the chemical vapor deposition method, and then immersed the sponge into the aqueous solution of a fluoropolymer. The acquired sponge material immediately absorbed heavy oil in the water as well as oil that was floating on the water's surface. Wei et al [24] prepared a quartz sand filter material that is superhydrophilic and underwater superoleophobic due to the chitosan modification method, with a contact angle of underwater oil as high as 153.2°. Biomass is often discarded or burned, which pollutes the environment and causes the waste of resources. Using biomass to prepare special wettability materials not only avoids the complicated process of traditional special wettability materials, but also introduced a new way of recycling waste biomass. The separated substrates mainly include meshes and sponges [25,29–31], but the mesh is easily fouled and blocked by oil adhesion. Later, in order to prevent membrane fouling, researchers developed superhydrophilic and superoleophobic membranes, but most of this technology needed special equipment, complicated technology processes, and large-scale implementation was difficult to control. The modified sponge treatment of oily wastewater possessed a simple operation, high separation efficiency, heavy oil concentration, etc. It is typically employed in emergency environmental protection accidents with large concentrations of oily wastewater, but it is not frequently used in the treatment of low-concentration industrial oily sewerage. Therefore, we proposed employing a quartz sand hard particle filter material as the separation material if the oil/water mixture is infiltrated by lyophobic liquid to remove water (or oil) by means of reverse extrusion. Therefore, changing the wettability of the quartz sand surface to make it superoleophobic underwater and superhydrophobic underoil was considered, which would be employed for dye adsorption and the separation of oil/water mixtures.

In this paper, to create quartz sand with underwater superoleophobic qualities and underoil extremely hydrophobic qualities, a corn cob was coated onto quartz sand. The angles at which the underwater oil and underoil water interacted with the quartz sand filter media covered in corn cob were, respectively, 150.3°~154.6° and 132.2°~154.6°. The filter material had a good capacity for adsorbing malachite green, and could successfully separate an oil/water mixture. In addition, we conducted studies on the synchronous separation of an oil/water mixture and dye adsorption. The quartz sand modified with the corn cob had strong mechanical wear resistance and good stability in harsh environments. Therefore, the quartz sand modified with the corn cob has great potential for oil removal and dye adsorption.

## 2. Materials and Methods

### 2.1. Materials

The quartz sand (0.3–0.6 mm) was obtained from Shimian Jinghong Mining Co., Ltd., while the corn cobs were derived from agricultural waste locally. The following chemicals were bought from Rionlon Bohua (Tianjin, China) Pharmaceutical and Chemical Co., Ltd., Tianjin, China: dichloromethane, anhydrous ethanol, trichloromethane, cyclohexane, petroleum ether, sodium hydroxide, and malachite green (MG). Engine oil was purchased from Sinopec Great Wall lubricating oil, and derv was obtained from a nearby petrol station. The supplier of rapeseed oil was Yihai Kerry Food Marketing Co., Ltd (Shanghai, China).

### 2.2. Preparation of Quartz Sand Filter Media with Special Wettability

In total, 4 g of corn cob of a 400 mesh (0.0385 mm) was added into 100 mL of 9% NaOH solution, stirred at 1600 r/min for 12 h, and then ultrasonic shocked for 2 h with an ultrasonic cleaner of 40 kHz. The mixture was then centrifuged at 4000 r/min for 3 min after being ultrasonically dispersed for 2 h. The corn cob was washed with distilled water

until the pH was 7. It was then dried at 60 °C to generate the activated corn cob. To create a uniform dipping solution, the activated corn cob was mixed with 100 mL of anhydrous alcohol, 2 g of waterborne polyurethane, and magnetic stirring for 4 h. In total, 20 g quartz sand was immersed into the dipping solution for 48 h, then screened out, and dried in an oven at 60 °C for 48 h. The result was a corn-cob-coated quartz sand with underwater superoleophobic qualities and underoil extremely hydrophobic qualities (hereafter referred to as CCQS).

### 2.3. Characterization of Filter Media

An infrared spectrometer was used to examine the chemical composition of the filter material (FTIR, VERTEX 70). An optical contact angle measuring device was utilized to measure the static contact angle (OCA25). A glass slide was covered with quartz sand, and water was then dumped into the quartz glass dish (or oil). In total, 5 μL of oil (or water) was then drip-fed. Three distinct locations recorded the underwater oil's (or underoil water's) contact angle 15 s later, and the average value was calculated. A low-vacuum scanning electron microscope was used to describe the morphology of the filter material (SEM, JSM-5600LV). A versatile electronic energy spectrometer was utilized to evaluate the elements (XPS, PHI-5702).

### 2.4. Separation of Oil/Water Mixture

Figure 1 showed the oil/water separation installation, which was composed of a 10 cm long filter column on both sides and a 20 cm long and 3 cm inner diameter T-shaped glass tube in the center. The interface was fastened with screws and joined with a flange. The quartz sand was intercepted and fixed using a 200 mesh steel mesh. The same volume of CCQS filter material was used to fill both filter columns.

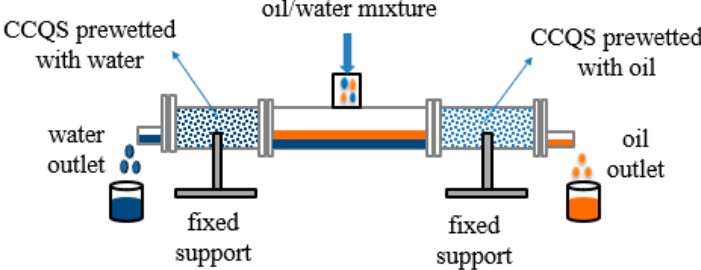

**Figure 1.** Schematic diagram of the experiment installation.

Oil and water were individually prewetted into the two filter columns. The upper mouth of the glass tube was used to inject the oil/water mixture (50% v:v). Gravity caused the stratification of water and oil. Because the filter media's wettability varied, the oil phase and water phase flowed out from both sides, respectively. As a consequence, oil/water separation was accomplished.

An instrument for measuring total carbon and total nitrogen (multi N/C 2100) was used to assess the amount of oil in the filtered water. The moisture meter was used to calculate the water in the oil (SN-WS200A). Formula (1) was used to determine the separation efficiency:

$$\eta = \frac{C_i - C_e}{C_i} \times 100\% \tag{1}$$

where $\eta$ is for the efficiency of oil (or water) separation; $C_i$ stands for the oil (or water) concentration before the separation process; and $C_e$ stands for the oil (or water) concentration after the separation process.

The filter columns' permeability coefficients were computed as follows:

$$K = \frac{Q \cdot L}{A \cdot \Delta h} \tag{2}$$

where $K$ is the permeability coefficient; $Q$ denotes the seepage discharge per unit time. $L$ is the thickness of the filter layer; $A$ denotes the cross-section area of the water; and $\Delta h$ denotes the filter layer's head loss.

The instrument was used to calculate the intrusion pressure of lyophobic liquid (Figure S1). A 10 mm internal diameter glass tube was filled with 5 cm thick CCQS to simulate the intrusion pressure of water. Then, prewetting was performed using oil. The lifting table's height was set such that the oil surface was the same height as the top surface of the CCQS. The aim was to keep the static pressure of the oil within and outside the glass tube constant. The glass tube was colored with methylene blue for easier viewing, and water was progressively added from the top using a rubber dropper until the water just brushed the surface of the CCQS. The incursion height $h_{max}$ was now the altitude difference between the surface of the CCQS and the water. The average values of three replicates were computed. To gauge the oil's incursion pressure, water was replaced for oil. The following formula was used to determine the intrusion pressure [32,33]:

$$P = \rho g h_{max} \tag{3}$$

where $P$ is the incursion pressure value; $\rho$ is the density of oil or water; $g$ is the gravitational acceleration; $h_{max}$ is the greatest altitude of oil or water.

### 2.5. Wetting Stability

We prepared 4 beakers filled with ethanol, then soaked the CCQS in the beakers, placed them in an ultrasonic cleaner, and then continuously ultrasonically vibrated them at 40 kHz for 4 h, 8 h, 16 h, and 24 h, respectively. In a 60 °C oven, the CCQS was removed to dry. The contact angles between underoil water (cyclohexane) and underwater oil (dichloromethane) were then calculated. The filter media's resistance to mechanical wear was examined. After being submerged in the aqueous solution for 24 h at pH levels of 1, 5, 7, 11, and 14, the CCQS was removed to dry in a 60 °C oven. The quartz sand's mechanical wear resistance was then assessed using the contact angle. The CCQS was placed in air for 30 d, 60 d, and 150 d, respectively, and the contact angle was measured to examine its resistance to air exposure.

### 2.6. Dye Adsorption

To create an MG stock solution, we dissolved 1 g MG in 1000 mL of distilled water. This MG solution was then deliquated with deionized water to the appropriate concentration. A specific amount of CCQS was introduced into a 50 mL centrifuge tube that already contained 30 mL of MG liquor. The centrifuge tube was then placed in a vibrator with constant temperature vapor bathing and shocked for a period of time at 25 °C at 165 r/min. Supernatant fluid was then obtained by centrifuging the sample for 5 min at 3500 r/min in a centrifuge. The MG concentration was determined using a UV spectrophotometer at $\lambda = 665$ nm. Next, the removal rate and adsorption capacity were computed. The following are the formulas used:

$$\eta_{MG} = \frac{C_{iMG} - C_{eMG}}{C_{iMG}} \times 100\% \tag{4}$$

$$q_e = \frac{C_{iMG} - C_{eMG}}{m} V \tag{5}$$

where $\eta_{MG}$ stands for the MG removal rate; $C_{iMG}$ and $C_{eMG}$ stand for the MG initial and equilibrium concentrations, respectively; $q_e$ stands for the MG adsorption capacity; $V$ stands for the volume of MG liquor; and $m$ stands for the amount of CCQS in grams.

### 2.7. Synchronous Separation of Oil/Water Mixture and Adsorption Dyes

The experimental device was the same as the one mentioned in Section 2.4. Petroleum ether and MG aqueous solutions of different concentrations were flowed into the liquid inlet chamber at a volume ratio of 1:1 to perform the synchronous separation of the oil/water

mixture and dye adsorption experiments. When oil, water or dye invade the lyophobic filter material layer, it can be removed by reverse extrusion. Three rounds of the procedure were required until the lyophobic liquid was no longer visible in the liquid poured for filtering.

## 3. Result and Discussion

### 3.1. Surface Wettability

In Figure 2, seven different types of oil static contact angles with the surface of the CCQS are displayed. Airborne water and oil droplets soon dispersed and completely soaked the quartz sand. The contact angle between the oil and water was 0°. The corn cob coating on the quartz sand had both superhydrophilic and superoleophilic properties in the air, because it included amphiphilic groups such as hydroxyl, carboxyl, etc. The underwater oil and underoil water contact angles of the unmodified quartz sand were 125° and 100°, respectively. As can be seen from Figure 2, the underwater oil and the underoil water contact angles with the CCQS were, respectively, 150.3~154.6° and 132.2~154.6°, achieving the level of underwater superoleophobic qualities and underoil extremely hydrophobic qualities.

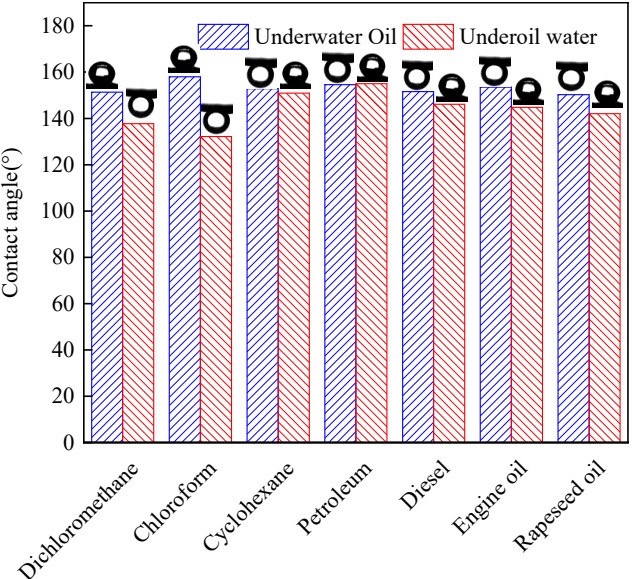

**Figure 2.** Contact angles of underoil water and underwater oil.

### 3.2. Wetting Stability

Stability is a crucial consideration when using ultra-wettable materials. The result of the stability is shown in Figure 3. The mechanical wear characteristics of the quartz sand could be assessed using ultrasonic oscillation [34]. Figure 3 shows that the underwater oil and underoil water contact angles of the CCQS were both above 150° after ultrasonic shocking of 4 h, 8 h, 16 h, and 24 h, stating that the CCQS had strong mechanical wear resistance. When the CCQS was soaked in a strong acid solution of pH = 1 and in a strong alkali solution of pH = 12, the contact angle of the CCQS underwater oil decreased from 151.3° to 147.9° and 150.9° to 148.4°, respectively. In other cases, the contact angles of the CCQS were both above 150°. When the CCQS was placed in air, the contact angle descended with the extension of time. At 150 d, the contact angles of the underwater oil and underoil water decreased from 151.3° to 138° and from 150.9° to 131.4°, respectively; the possible reason is that the grafted corn cob on the surface of the CCQS fell off a little. The study demonstrated that the CCQS had a strong ability to resist harsh environments [24].

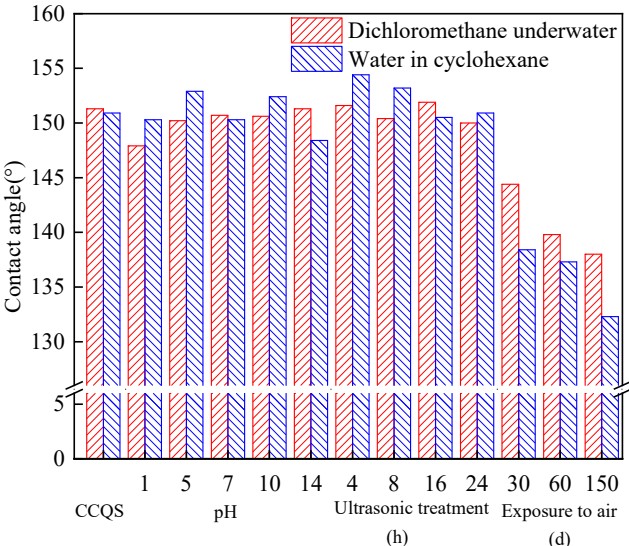

**Figure 3.** The contact angle of CCQS after ultrasonic treatment, soaking in acid–alkali solutions and air exposure.

### 3.3. Characterization of Filter Media

### 3.3.1. FTIR Analysis

Figure 4 presented the results of the quartz sand infrared spectrum analysis. A strong, wide absorption peak appeared between 3710 and 3115 cm$^{-1}$, which were the stretching vibration peaks of the -OH group from cellulose. Compared to the unmodified quartz sand, the CCQS was more hydrophilic and had a lot of hydroxyl groups. The modified quartz sand's infrared spectra revealed a new absorption peak at a wavelength of 1627 cm$^{-1}$. This peak was related to the C=O group and it was the characteristic absorption peak of lignin [35]. The smaller absorption peak at approximately 1533 cm$^{-1}$ was the vibration of the aromatic skeleton of the C=C group, and the absorption peak between 1364 cm$^{-1}$ was the stretching vibration in the plane of the C-H bond of the aromatic ring [36]. The stretching vibration of the C-O molecule was at its absorption peak between 1300 cm$^{-1}$ and 1000 cm$^{-1}$. The above results of infrared spectroscopy proved that the corn cob had been successfully adhered onto the quartz sand. The chemical composition of the corn cob included many hydrophilic polar functional groups, which played an important role for the corn-cob-modified quartz sand to achieve water/oil emulsion separation.

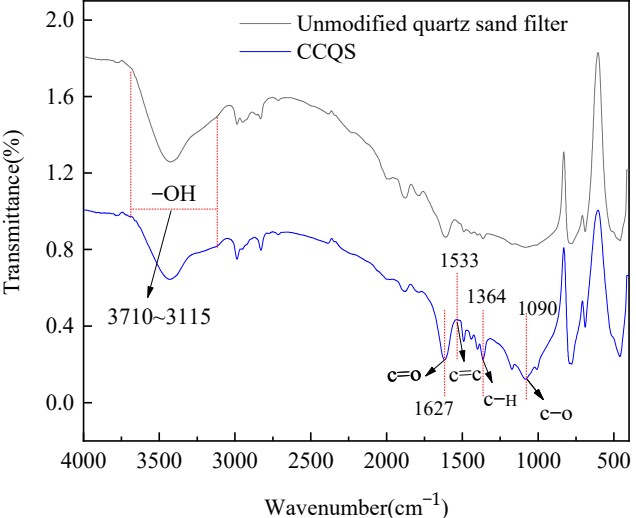

**Figure 4.** FTIR spectra of CCQS and unmodified quartz sand filter.

### 3.3.2. SEM Analysis

The CCQS and unmodified quartz sand SEM images (Figure S2) showed that the unmodified quartz sand (Figure S2a–c) was relatively smooth and clean. The CCQS (Figure S2d–f) had many agglomerates on the surface, which was waterborne polyurethane that adhered to the corn cob particles on the quartz sand. The surface of the CCQS exhibited a rough micro/nano structure, as seen in the high-magnification SEM picture. It revealed that the CCQS surface had a rougher structure than the unmodified quartz sand, which enhanced the wettability of the CCQS.

### 3.3.3. XPS Analysis

The elements of the modified and unmodified quartz sand were analyzed using XPS. The XPS wide-scan survey spectrum (Figure S3) confirmed that the CCQS contained a trace amount of Na, which should have been introduced when the corn cob was activated by NaOH, and manifested once the corn cob was successfully adhered to the quartz sand. The presence of a feeble N1s peak at 398.4 eV demonstrated that the quartz sand was adhered to the waterborne polyurethane. High-resolution XPS scans of the spectrum of the Si2p spectral region before and after modification appear in Figure 5a,b. The Si2p XPS spectra of unmodified quartz sand fitted well into the two peaks at 103.0 eV and 103.3 eV (Figure 5a), with the proportions of 47.34% and 52.66%, respectively, belonging to Si-OH and $SiO_2$ [37], demonstrating that the quartz sand contained hydroxyl on the surface before modification. For the CCQS, the Si2p XPS spectra (Figure 5b) decomposed into three peaks at 103.0 eV (Si-OH), 103.2 eV ($SiO_2$), and 102.2 eV (Si-O-C) [38], accounting for 36.12%, 42.77%, and 21.11%, respectively.

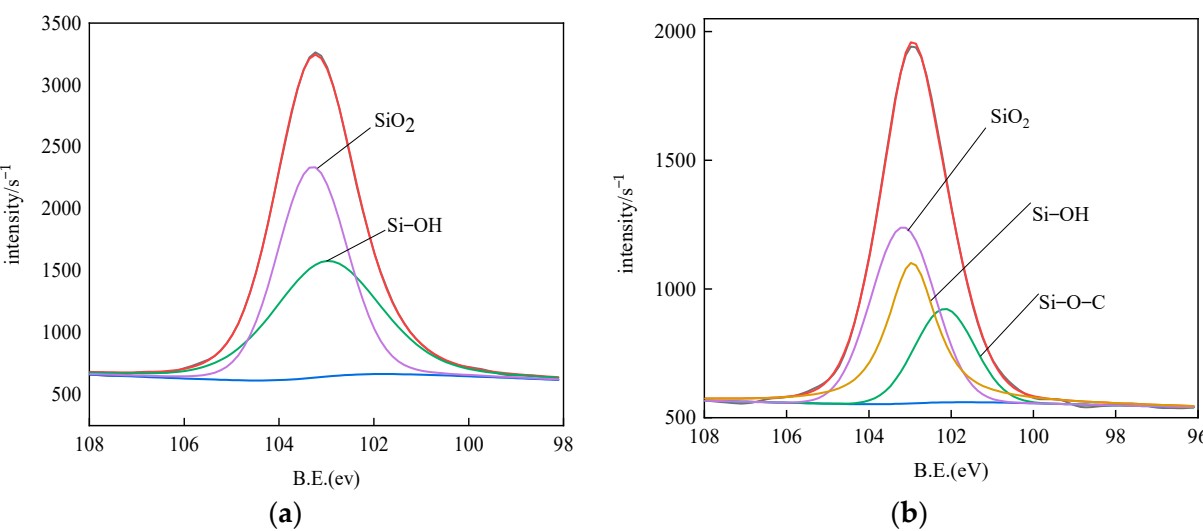

**Figure 5.** Deconvolution of XPS Si2p high-resolution spectra of quartz sand before and after modification. (**a**) Unmodified quartz sand; (**b**) CCQS.

### 3.4. Separation of Oil/Water Mixture

Figure S4 displays the separation device for the oil and water mixture. The superoleophilic and underwater extremely hydrophobic quartz sand was on the right side, while the superhydrophilic and superoleophobic quartz sand was on the left side. Figure 6 displays the removal efficiencies of various types of oil/water mixtures. The removal rate of water and oil reached above 99.93%, with the CCQS manifesting an excellent separation efficiency. After 10 test cycles, the removal effect was still up to 99.00%, confirming that the CCQS had a good oil/water separation performance. The extremely high separation efficiency of the oil/water mixture was mainly attributed to the surface wettability and capillary pressure of the CCQS, as shown in Figure S6. When the CCQS was prewetted with water, it exhibited underwater superoleophobic properties, and the capillary pressure

of the oil pointed to the water inlet side, preventing the passage of oil. Similarly, when the CCQS was prewetted with oil, it showing extremely hydrophobic properties underoil, thus, preventing the passage of water.

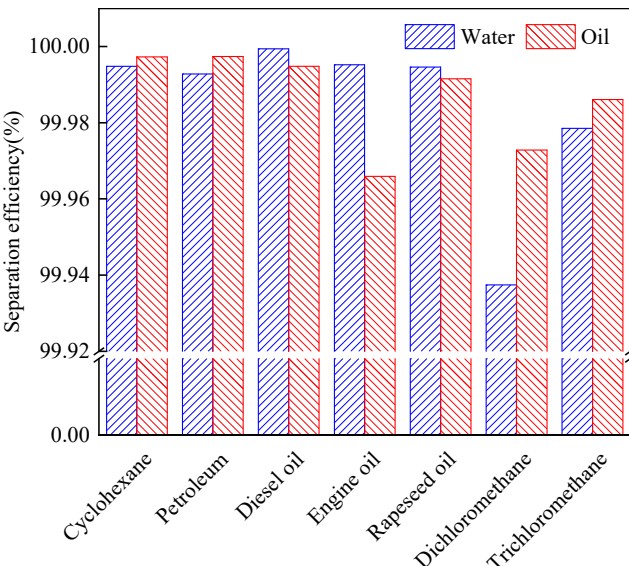

**Figure 6.** Separation efficiency of different kinds of oil/water mixture.

When the corn cob was activated by NaOH, more hydrogen bonds were released and grafted onto the quartz sand surface. Hydrogen bonds had a strong adsorption effect on MG, and, as a result, the CCQS could adsorb MG efficiently.

The permeability coefficient is linked to the type, viscosity, and density of oil [39]. Figure S5 exhibits the permeability coefficient of oil and water, which was used to measure the separation speed. The detailed calculation process of the permeability coefficient can be found in reference [40]. Rapeseed oil, diesel oil, and engine oil all had high viscoelastic properties. As a result, the oil production rate of the oleophilic side of the filter column was slow, and the permeability coefficients were 0.13, 0.47, and 2.68 m/h, respectively. The viscosity of cyclohexane, petroleum ether, dichloromethane, and trichloromethane was small, so the oil production rate of the oleophilic side was quick, and the permeability coefficients were 7.76 m/h, 8.06 m/h, 12.00 m/h, and 13.00 m/h, respectively. Water had a 6.84 m/h permeability coefficient. Therefore, for oil with reduced viscosity, the CCQS had a higher separation speed.

Figure 7 depicts the measured CCQS incursion pressure. A higher incursion pressure meant that the apparatus was more stable. The superhydrophilic and superoleophilic sides, respectively, were the incursion pressures of underoil and underwater. Figure 7 shows that the experimental and theoretical values of the intrusion pressure of seven oil/water mixtures were slightly different. Among them, the theoretical value of the underoil water of engine oil was the smallest, which was only 0.07 kPa, while the experimental intrusion pressure was the largest, reaching 0.44 kPa. The experimental intrusion pressure of the underwater oil of engine oil was the largest, which was 0.22 kPa, while the theoretical intrusion pressure was the smallest, which was only 0.078 kPa. This difference resulted from the engine oil's very high viscosity and slow flow rate [40].

The capillary pressure diagram can be seen in Figure S6. The theoretical incursion pressure formula based on Laplace's theory was [41,42]:

$$\Delta P = \frac{2r \cos \theta}{R} \tag{6}$$

where $\Delta P$ is the theoretical intrusion pressure; $r$ is the liquid's surface tension; $\theta$ is the liquid's contact angle with the surface preparation; $R$ is the radius of the capillary tube in the filter layer.

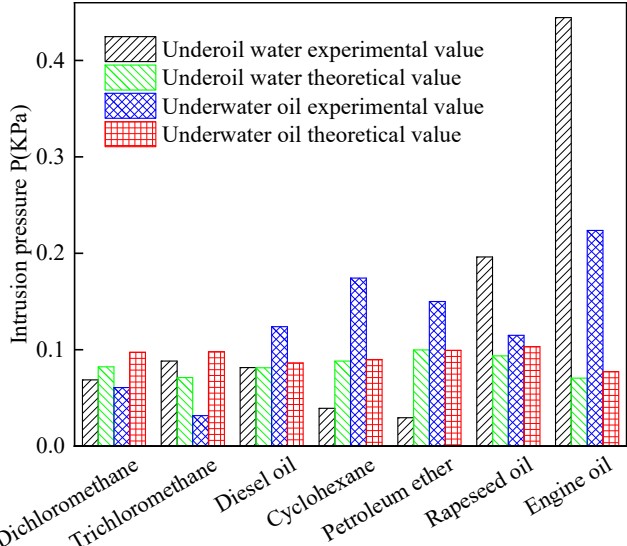

**Figure 7.** Intrusion pressure of CCQS.

The superhydrophilic filter media had a water contact angle of 0° in air, and the water's capillary pressure was pointed toward the output side, causing the water phase to flow through. The superoleophilic filter media's oil contact angle was also 0°, and the oil phase was forced to exit through the filter media's gap due to the oil's capillary pressure. However, the underwater superoleophobic surface's (underwater superhydrophobic surface's) underwater oil contact angle was larger than 90° underwater (or underoil). Therefore, $\Delta P$ was over 0°, and the oil's (or water's) capillary pressure was directed toward the intake side, preventing the oil phase (or water phase) from flowing through the filter media and efficiently separating the oil/water mixture [32,33]. Furthermore, this gadget may be employed to overcome oil density-limiting and liquid accumulation issues.

*3.5. Dye Adsorption*

3.5.1. Effect of Adsorbent Dosage

The dosage of quartz sand on the dye adsorption effect is presented in Figure 8a. As the quartz sand dosage rose from 0.5 g to 3 g, the removal rate continued to increase from 55.47% to 99.15%, and the adsorption capacity decreased from 5.51 mg/g to 1.65 mg/g. When the mass of quartz sand continuously grew from 3 g to 20 g, the removal rate of the dye remained basically unchanged, and the adsorption capacity gradually decreased from 1.65 mg/g to 0.25 mg/g. Because of the electrostatic attraction between the negatively charged CCQS and cationic dye molecules, the dye was adsorbed [43]. With the increase in quartz sand dosage, the total active sites bound to MG molecules increased, resulting in a decrease in the concentration of dye solution after adsorption, thus, increased the removal rate. However, with the increase in quartz sand dosage, the dye solute remained unchanged. When the adsorption equilibrium of quartz sand was not reached, the solute molecules absorbed by the unit mass of quartz sand were declined, and the adsorption capacity decreased continuously.

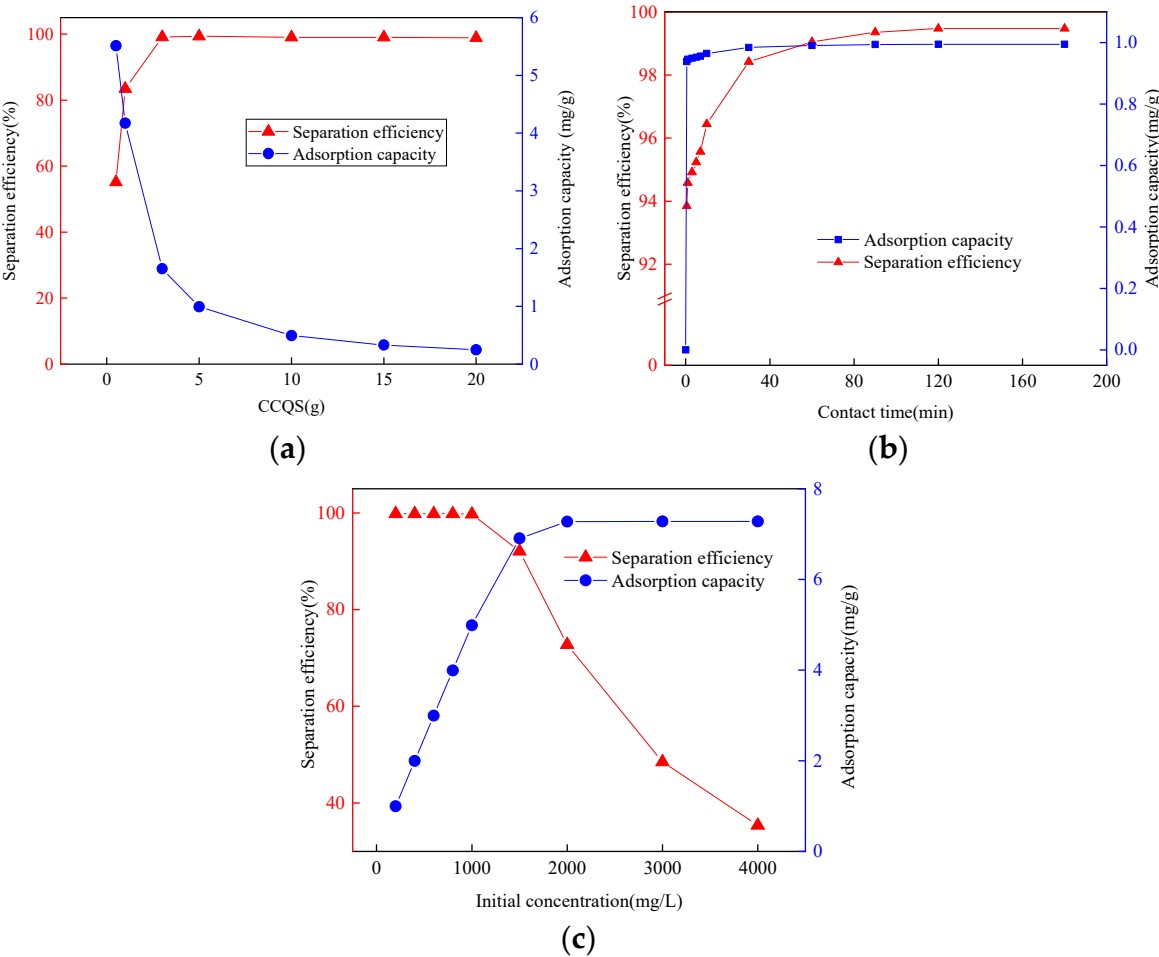

**Figure 8.** (**a**) Effect of CCQS dosage on adsorption capacity and separation effectiveness. Initial MG concentration was 200 mg/L and adsorption time was 1 h. (**b**) Effect of contact time on quartz sand capacity for adsorption. Initial MG concentration was 200 mg/L, and the dosage of CCQS was 5 g. (**c**) The effect of starting concentration on the rate of MG removal and adsorption capacity. The CCQS dosage was 5 g, and the adsorption time was 1 h.

### 3.5.2. Effect of Adsorption Time

The effect of the adsorption time on the dye removal rate change is shown in Figure 8b. In the first 5 min of the reaction, the adsorption rate was the fastest, with the removal rate reaching 95.24%. When the adsorption time increased from 5 min to 30 min, the removal rate gradually increased to 98.42%, and at 120 min finally reached equilibrium, with the highest removal rate of 99.47%. The reason was that at the beginning of adsorption, there were several adsorption locations on the CCQS surface, so MG was immediately adsorbed. The adsorption capacity of the CCQS steadily reduced as the adsorption time increased and achieved an equilibrium. The nonlinear fitting of the adsorption kinetics of CCQS on MG (Figure S7) confirmed that the $R^2$ of quasi-first-order kinetics and quasi-second-order kinetics was 0.9956 and 0.9972, respectively. The adsorption of the CCQS on MG fitted well with the quasi-second-order kinetics model, manifesting that the adsorption process was mainly affected by chemical reactions [44,45].

### 3.5.3. Effect of Initial Dye Concentration

The effect of the initial dye concentration on the dye removal rate change was is in Figure 8c. The dye removal rate maintained above 99% and the adsorption capacity rose from 0.99 mg/g to 4.99 mg/g as the dye concentration increased from 200 mg/L to 1000 mg/L, revealing a linear upward trend. The dye removal rate declined from 99.74 %

to 92.06 % when the concentration was raised from 1000 mg/L to 1500 mg/g, although the adsorption capacity rose from 4.99 mg/g to 6.90 mg/g. As the initial concentration of dye continued to increase, the removal rate continuously decreased. The dye removal rate was reduced to 35.39% at a starting concentration of 4000 mg/L, and the saturated adsorption capacity of quartz sand stabilized at approximately 7.28 mg/g. As the initial concentration increased, the concentration gradient increased, which caused the adsorption kinetics to increase, and the adsorption capacity also increased. Therefore, the CCQS could quickly and successfully absorb MG and separate aqueous cationic dyes. The nonlinear fitting of the CCQS of MG adsorption isotherms (Figure S8) showed that the $R^2$ values of Langmuir and Freundlich isotherm adsorption models were 0.9897 and 0.8622, respectively. The adsorption of the CCQS on MG fitted well with the Freundlich isotherm adsorption model, demonstrating that the adsorption process was mainly affected by the multimolecular layer adsorption [44,45].

### 3.6. Synchronous Separation of Oil/Water Mixture and Adsorption Dye

The effect of the synchronous separation of the oil/water mixture and dye adsorption is displayed in Figure 9. The particle size of the CCQS was 0.18~0.42 mm and the initial concentration of MG was 100 mg/L. The removal efficiency of oil in the separation mixture was always above 99.99%. Additionally, when the separation volume increased to 0.8 L, the removal efficiency of water declined from 99.99% to 97.56%. The filtered oil gradually penetrated into the water phase part, resulting in a sudden drop in the water separation efficiency (the prominent point 0.8 L, as shown in Figure 9). Reverse extrusion was used at this point to remove the water phase from the filter media's pores, and filtering was resumed. The efficiency of water removal was increased to 99%. This was because the height of the water in the middle inlet cavity was too high to reach the invasion pressure, resulting in water penetrating through the filter column and the removal rate of water decreasing. When the filtration volume increased from 3.6 L to 4 L, the dye removal rate decreased from 97.79% to 87.71%, then continued to filter, and the removal rate rapidly declined. This was because, in the early stage, the saturated adsorption capacity of quartz sand was not reached, so it had a high adsorption efficiency. The adsorption sites of quartz sand were constantly reduced when the separation volume increased, resulted in the continuous decline of the dye removal rate.

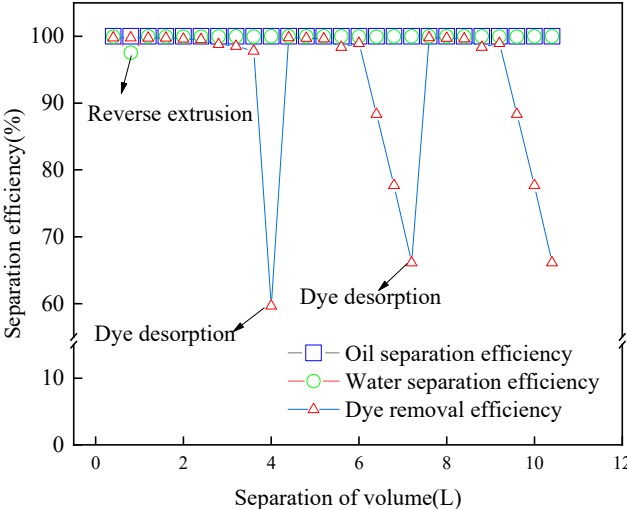

**Figure 9.** Synchronous separation of oil/water mixture and dye removal effect.

### 3.7. Modified Mechanism

The corn cob is composed of cellulose. The general formula of cellulose is $(C_6H_{10}O_5)n$ [46], which is an extremely regular linear polymer formed by linking glucoside repeating units through β-1, 4-glycosidic bonds. The special morphology and supramolecular structure

of cellulose causes its extremely reactive hydrogen bonds to lock in the crystallization zone, making it challenging to interact with reagents and difficult to carry out all kinds of cellulose chemical reactions [47]. Alkali activation was utilized in this investigation to increase the solubility of cellulose by reducing its crystallinity and exposing a significant amount of hydroxyl groups [48,49]. When the activated cellulose was in contact with quartz sand, the -CH$_2$OH group on cellulose reacted with -OH of the quartz sand [24,50], and the hydrophilic group became grafted to the quartz sand, as depicted in Figure 10. The roughness was increased by the addition of waterborne polyurethane, which strengthened the bond between the cellulose and quartz sand.

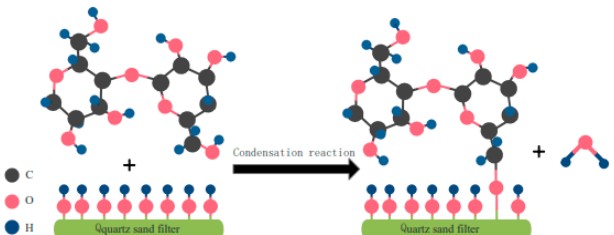

**Figure 10.** Schematic diagram of preparing CCQS.

## 4. Conclusions

By using a straightforward dip-coating technique, we were able to successfully manufacture quartz sand filter media modified with a corn cob that had underwater superoleophobic qualities and underoil extremely hydrophobic qualities. The contact angles between the underwater oil and the underoil water with the CCQS were, respectively, 150.3~154.6° and 132.2~154.6°. The FTIR, SEM, and XPS analyses findings showed that the surface of quartz sand was successfully grafted with a corn cob. The devised successive oil/water separation device exhibited remarkable separation effects. The oil/water mixture separation efficiencies were more than 99.93%, and after 10 cycles of testing, the separation efficiencies were still up to 99.00%. The CCQS could effectively adsorb MG with a removal rate of 99.73% and with an adsorption capacity of 7.28 mg/g. The corrosion resistance, mechanical durability, and environmental durability of the CCQS were all outstanding. The CCQS was capable of constantly and effectively separating the oil/water mixture. Synchronously, the CCQS successfully adsorbed dyes in wastewater. As a result, the CCQS has potential for industrial applications.

**Supplementary Materials:** The following supporting information can be downloaded at: https://www.mdpi.com/article/10.3390/su14169860/s1, Figure S1: Diagram for intrusion pressure device. Figure S2: SEM images of (a–c) unmodified quartz sand; (d–f) CCQS. Figure S3: XPS spectra of Filter material. Figure S4: Separation device for oil and water mixture. Figure S5: The permeability coefficients of the water outlet and oil outlet when filtering different oils. Figure S6: Theoretical wetting model of superhydrophilic and underwater superoleophobic or superoleophilic and underoil superhydrophobic filter media. (a) In air, water can pass through superhydrophilic quartz sand; (b) in air, oil can pass through superoleophilic quartz sand; (c) underwater, oil cannot pass through underwater superhydrophobic quartz sand; (d) under oil, water cannot pass through underoil superhydrophobic quartz sand. Figure S7: Adsorption Kinetic fitting. Figure S8: Adsorption isotherm fitting.

**Author Contributions:** Conceptualization, W.B.; Investigation, G.H. and M.S.; Methodology, G.H.; Supervision, W.B.; Writing—original draft, C.Y.; Writing—review and editing, Z.T. All authors have read and agreed to the published version of the manuscript.

**Funding:** The National Natural Science Foundation of China (nos. 51668032 and 52060014); The Department of Education of Gansu Province: The Excellent Postgraduate Student "Innovation star" Project (nos. 2022CXZX-558 and 2022CXZX-561).

**Institutional Review Board Statement:** Not applicable.

**Informed Consent Statement:** Not applicable.

**Conflicts of Interest:** The authors declare no conflict of interest.

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
