# Peer review of "Preparation of Special Wettability Quartz Sand Filter Media and Its Synchronous Oil/Water Mixture Separation and Dye Adsorption"

_sustainability, doi:10.3390/su14169860_

Round 1

Reviewer 1 Report

This study reports the fabrication of corn cob-covered quartz sand (CCQS) as a filter media for oil/water separation and dye adsorption. The surface chemical composition and wettability of the obtained CCQS were characterized. The separation performance was tested using an oil/water mixture and dye solution. The research topic is of interest, and the performance of this material is good. However, there are some issues with their experimental design and the English language. Here are my comments.

1.      The authors should report the size and size distribution of the quartz sand.

2.      The coating amount of the corn cob on the quartz sand surface was not reported. The author might also need to optimize the coating amount.

3.      The separation experiments were driven by gravity, which means the flux would decline as the solution level goes down. Therefore, more details about the calculation of permeability should be given. Besides, since the flux is always changing, how did the authors compare the permeability coefficients in a fair way?

4.      In the calculation of Laplace pressure, R is the radius of the capillary tube or the equivalent radius of the porous media. What is the “mesh radius” here (Line 294)?

5.      When exposed to air, the CCQS showed reduced contact angles. What are the possible reasons?

Reviewer 2 Report

Dear Editor

Thank you for the invitation to review the manuscript entitles "Preparation of special wettability quartz sand filter media and its synchronous oil/water mixture separation and dye adsorption". The manuscript matches the scope of the journal and describes the Corn cob-covered quartz sand filter media (CCQS) was fabricated by grafting corn cob onto the surface of quartz sand using the dip-coating technique. The graft material was used for separation of oil/water mixtures and adsorbed dyes.

The general idea of the paper seems to be good. However, there are several major technical challenges that should be effectively addressed.

 Best Regards

Comments to the authors:

1.      The abstract has been briefly written and should be enriched by adding the main ideas and contributions.

2.      Line 17, please replace MB by MG

3.      There are some grammatical errors and typos that should be corrected before publication

4.      In the introduction section the authors did not present the drawbacks and gaps of literature with the current work, and particularly, how the proposed approach aims at filling these gaps.

5.      Line 97, "Once the solution's pH reached 7, the solution was washed with distilled water" how the authors washed the solution!!!

6.      Line 111, FTIR should be mentioned before CA and SEM, because the FTIR observation is confirmed the grafting of the material.

7.      Caption of Figure 1 should be schematic diagram…..

8.      Figure 2 not clear

9.      Line 222, 3.3.2. FTIR Analysis should be move to be the first section in the results and discussion

10.  Line 216, the authors simply describe their observation in section 3.3.1. SEM Analysis without discussion

11.  The caption of Figure 4 wrong, it is not only for CCQS, please revise it?

12.  Results and discussion-Here, the authors simply describe their observation, but interpretation of facts the reader is not enough to find in all sections. This part of the MS should be enriched with more and deep interpretation.

Round 2

Reviewer 1 Report

My concerns have been well addressed. The manuscript can be accepted.

Reviewer 2 Report

Dear Editor

I would like to inform you that the authors answer all of my comments, therefore, I recommend to accept in sustainability

Best Regards